# ProAKAP4 as Indicator of Long-Lasting Motility Marker in Post-Thaw Conditions in Stallions

**DOI:** 10.3390/ani14091264

**Published:** 2024-04-23

**Authors:** Marta Dordas-Perpinyà, Iván Yánez-Ortiz, Nicolas Sergeant, Vincent Mevel, Jaime Catalán, Jean-François Bruyas, Jordi Miró, Lamia Briand-Amirat

**Affiliations:** 1Oniris, Nantes Veterinary College, Cedex 03, 44307 Nantes, France; marta.dordas-perpinya@vet-alfort.fr (M.D.-P.); vincent.mevel@oniris-nantes.fr (V.M.); jfbruyas691@gmail.com (J.-F.B.); 2Equine Reproduction Service, Department of Animal Medicine and Surgery, Faculty of Veterinary Sciences, Autonomous University of Barcelona, E-08193 Cerdanyola del Vallès, Spain; ivan.yanez22@gmail.com (I.Y.-O.); dr.jcatalan@gmail.com (J.C.); 3UVSQ, INRAE, BREED, Université Paris-Saclay, 78352 Jouy-en-Josas, France; 4Ecole Nationale Vétérinaire d’Alfort, BREED, 94700 Maisons-Alfort, France; 5School of Veterinary Medicine, Faculty of Medicine, Health and Life Sciences, International University of Ecuador, Quito 170901, Ecuador; 6INSERM, UMRS, University of Lille, 59000 Lille, France; nsergeant@4biodx.com; 7SPQI, 4bioDx—Breeding Section, 59000 Lille, France

**Keywords:** ProAKAP4, AKAP4, stallion, spermatozoa, motility, mitochondrial activity

## Abstract

**Simple Summary:**

ProAKAP4, as a AKAP4 precursor, is the main protein of the spermatozoa flagellum in mammals. AKAP4, A-kinase anchor protein, is involved in sperm motility. The objective of the present study was to evaluate the evolution of proAKAP4 in horse semen post-thawing conditions during 3 h and its relationship with motility descriptors and mitochondrial membrane potential in the same conditions. The evolution in post-thawing conditions is the same either in proAKAP4 and motility parameters which gather that proAKAP4 can predict the evolution of the motility over the time.

**Abstract:**

ProAKAP4, a precursor of AKAP4 (A-kinase anchor protein) found in the flagellum of mammalian and non-mammalian spermatozoa, serves as a structural protein with established correlations to motility parameters across diverse species. This study aimed to determine the proAKAP4 level evolution in thawed stallion semen over a 3 h period, examining its correlation with motility descriptors and mitochondrial membrane potential. Utilizing sixteen ejaculates from four French warmblood stallions, this study involved maintaining thawed samples at 37 °C for 3 h, conducting proAKAP4 enzyme-linked immunosorbent assays (ELISA), computer-assisted sperm analysis (CASA), and mitochondrial membrane potential by JC-1 probe and flow cytometry at 0, 1, and 3 h post-thawing. The findings indicate significant positive correlations (*p* ≤ 0.05) between proAKAP4 levels and sperm total or progressive motility at all time points analyzed. Spermatozoa velocity descriptors (VAP, VCL, VSL) and spermatozoa lateral head displacement (ALH) display positive correlations (*p* ≤ 0.05) with ProAKAP4 at the 0 h post-thawing. ProAKAP4 concentration exhibits no discernible difference between batches with or without a cryoprotectant. Notably, proAKAP4 consumption remains insignificant within the initial hour after thawing but becomes significant (*p* ≤ 0.05) between 1 and 3 h post-thawing. In summary, proAKAP4 demonstrates positive correlations with total and progressive motility in stallion semen for up to 3 h after thawing, albeit showing a noticeable decrease starting from the first hour post-thawing, indicating a progressive consumption as a result of spermatozoa motile activity.

## 1. Introduction

The cryopreservation of sperm over extended periods is pivotal for the survival, maintenance, and enhancement of genetic diversity and improvement within species and breeds worldwide [1]. In the equine field, semen cryopreservation has been important for different reasons like spreading good genetics worldwide, minimizing disease transmission, producing offspring after the death of the stallion [2] or keeping rare genetic lines from endangered breeds [1].

The most important point about frozen-thawed semen is to maintain the capacity of fertilization after thawing. One of the major impediments to the development of the horse frozen semen industry is the lower fertility with frozen semen versus to cooled semen [3]. However, Lomis (2001) [3] concluded that acceptable fertility with frozen semen is possible with good quality semen. A good quality semen once thawed will reach the fertilization site to penetrate the oocyte [1]. Thawed semen is considered good when its progressive motility is over 35%, [4] but this is not a warranty that the semen will be fertile [4,5]. The definition of semen fertility is not easy and precise [5], it is usually related to the seminal quality parameters [4,6]. The estimate of semen fertility has been always a challenge, frozen semen appears to have shorter longevity once thawed [2] and lower conception rates [7,8]. Insemination of the mare with frozen semen is a challenge due to the short lifetime of the spermatozoa once thawed. It is also difficult to predict how long the semen will be alive and motile after thawing to adapt to the ovarian scanning routine of the mare [8] for artificial insemination with frozen-thawed semen. The most commonly employed protocol to inseminate a mare with frozen semen to obtain a high pregnancy rate, the mare has to be inseminated within a 6 h window around ovulation [2,9]. To reach this objective the mare has to be scanned regularly to catch the ovulation time.

The traditional semen analysis (sperm concentration, subjective motility and sperm morphoabnormalities) has a reduced predictive value in the evaluation of stallion fertility. An important number of sperm tests (e.g., sperm head morphometry, the hypoosmotic swelling test, acrosome integrity, membrane integrity, progesterone receptor exposure, DNA integrity or fragmentation, intracellular calcium, intracellular ROS, and mitochondrial activity) have been developed in order to increase the sperm fertility prediction. However, most examine only a single or a narrow range of the attributes that a sperm must possess. Flow cytometry offers the possibility of quickly and simultaneously combining many of these analyses [6]. On the other hand, the CASA-Mot (computer-assisted semen analysis for motility) can be used to rapidly assess total and progressive motility, as well as determine values for other kinematic variables [10]. Nowadays, the prediction of seminal quality has been usually related to motility parameters analyzed by CASA systems, flow cytometry analysis and formulas combining both [1]. However, semen quality is not always correlated with good fertility rates [11]. From a practical point of view, the search for new fertility predictors is an important field of research.

ProAKAP4 is a marker of sperm quality and male fertility in different species [12], including in stallions [13,14,15,16,17]. ProAKAP4 is the precursor of A-kinase anchor protein 4, which is the main protein of the fibrous sheath of the principal piece of the flagellum [12,18]. As an important part of the flagellum [19,20], proAKAP4 is a good marker of the post-meiotic stage of spermatogenesis [12,21,22] as well as a good marker of sperm progressive and total motility in stallions [13,14,15,16] and in other mammals like bulls [23], jackass [24], dogs [25], rabbits [26], rams [27] and dromedary [28]. In stallions, proAKAP4 has been established as a good marker of motility descriptors [13,14,15,16,17], since more deeper, rapid spermatozoa show a greater concentration of proAKAP4 [14] and ejaculates with a high percentage of slow sperm subpopulations show a lower concentration of proAKAP4 [16]. To sum up, proAKAP4 in stallions describes the motility and velocity of spermatozoa but not how the trajectory of this spermatozoa [16]. In humans, research with proAKAP4 has been deeper in terms of aspects of fertility. The main results show that proAKAP4 levels are low in asthenozoospermia, idiopatic fertility, associated with a low motility register and high ROS concentration, and in cigarettes smokers [12]. ProAKAP4 could be a good parameter to elucidate why motility parameters are not always correlate with a fertility outcomes.

However, proAKAP4 is not correlated with viability as not all viable spermatozoa are stained by proAKAP4 in stallion sperm after thawing [29]. ProAKAP4 is detected in living spermatozoa but proAKAP4 is degraded with cell death [22]. This study is complementary to previous research of the same research unit aiming to figure out proAKAP4’s role as a sperm quality marker. The precise aim is to correlate motility descriptors and mitochondrial membrane potential with proAKAP4 concentration in stallion frozen-thawed semen and explore its evolution over three hours to assess the potential of proAKAP4 as a long-lasting motility indicator. Another aim of this study is to test how a cryoprotectant affects proAKAP4 concentration and evolution over 3 h in post-thawing conditions.

## 2. Materials and Methods

### 2.1. Semen Samples

Sixteen ejaculates obtained from four French warmblood stallions, i.e., 4 ejaculates from each stallion, housed at Haras de Hus in Petit Mars, France, were utilized for this study, intended for both reproductive and sport (showjumping and dressage) purposes. The stallions ranged in age from 3 to 8 years at the time of semen collection. These animals were selected based on their proven fertility, excellent physical condition, and negative testing for Equine Arteritis, Infectious Anemia, and Contagious Metritis, conforming to CEE health requirements prior to semen production. Each stallion provided four ejaculates, from which three straws were extracted for further analysis.

Cryopreservation of all ejaculates was conducted between 2012 and 2016 at Haras de Hus, a collection center approved by European authorities. Consistent cryopreservation protocols were followed, employing INRAFreeze^®^ (IMV Technologies, L’Aigle, France) as an extender and MiniDigitcool^®^ (IMV Technologies, L’Aigle, France) as the automatic freezer.

### 2.2. Experimental Groups

Three straws for each ejaculate were thawed at 37 °C in a water bath for 50 s. Semen samples were deposited in an Eppendorf Tube containing 500 µL of INRA96^®^ previously warmed at 37 °C. The resulting mix was gently shaken, split into two aliquots. One aliquot was centrifuged at 600× *g* for 10 min to remove the cryoprotectant, the other one was not centrifuged. After centrifugation, the pellet was resuspended with 500 µL of INRA96^®^ pre-warmed at 37 °C. Then, both aliquots were kept inside an incubator at 37 °C. The analyses were performed at 0 h, 1 h and 3 h after thawing.

Semen analysis encompassed the evaluation of kinematic parameters utilizing computer-assisted semen analysis (CASA) (Section 2.3), determination of proAKAP4 concentrations (Section 2.4), and assessment of mitochondrial membrane potential using the JC-1 stain (Section 2.5). The semen concentration of each sample was calculated employing CASA methodology.

### 2.3. Motility Analysis

Sperm motility assessment was conducted utilizing a computer-assisted sperm analysis (CASA) system (IVOS II ^®^; Hamilton Thorne; IMV Technologies, L’Aigle, France). Post-thawing, all samples were maintained at 37 °C in a water bath for the initial minute, followed by transfer to a 37 °C incubator for the subsequent 3 h to sustain optimal temperature conditions for analysis. For examination, 3 μL of sperm samples were dispensed onto a prewarmed Leja^®^ slide at 37 °C. A total of 3 distinct fields and at least 1000 spermatozoa were examined to assess various motility descriptors including total motility (TM, %), progressive motility (PM, %), curvilinear velocity (VCL, μm/s), straight-line velocity (VSL, μm/s), average path velocity (VAP, μm/s), linearity (LIN, %), straightness (STR, %), amplitude of lateral head displacement (ALH, μm), and frequency of head displacement (BCF, Hz).

The CASA parameters were set as follows: magnification of ×200 with 10 randomly selected fields; acquisition of 60 images per second; capture of 30 sequences in each analysis; particle area set to 3 pixels; connectivity threshold set at 6; minimum number of images required to calculate ALH set at 10. The cut-off value for motile spermatozoa was established as VAP ≥ 20 μm/s, while progressive motility was defined as STR ≥ 80% and VAP ≥ 30 μm/s.

### 2.4. ProAKAP4 ELISA

ProAKAP4 quantification was performed using a commercially available quantitative enzyme-linked immunosorbent assay (ELISA). The analysis was conducted at the ONIRIS Laboratory in Nantes, France, utilizing the Horse 4MID^®^ kit (SPQI, Lille, France). This kit employs a quantitative ELISA sandwich assay designed to specifically detect and quantify proAKAP4 in equine semen samples. A reference curve was generated using a provided seven-point standard solution to determine the concentration of proAKAP4 present in each stallion’s thawed semen sample.

Sample preparation followed the manufacturer’s instructions for cryopreserved stallion semen. Forty microliters of diluted semen from each experimental group described in Section 2.2 were mixed with 160 µL of specific lysis buffer included in the kit. After vigorous vortex mixing for one minute, 200 µL of dilution buffer was added and rapidly mixed before loading into the 96-well coated plate. The plate was then incubated for 2 h under gentle agitation with an ELISA-plate horizontal shaker.

Following three successive washing steps, horseradish-conjugated detection antibody was added to each well and further incubated for 1 h. Subsequently, TMB substrate was added, initiating a color reaction after 10 min of incubation. The intensity of the color reaction, measured spectrophotometrically with a 450 nm filter, is quantitatively proportional to the amount of proAKAP4 present in each sample. The color reaction was halted by adding a stop solution to each well, and color intensities were measured using a spectrophotometer (Agilent BioTek 800TS, El Cajon, CA, USA).

Optical densities were then used to determine proAKAP4 concentration with the calculation sheet provided by the manufacturer (SPQI, Lille, France). Specifically, proAKAP4 concentration was expressed in ng per 10 million spermatozoa. This concentration was calculated using the formula: proAKAP4 ng/10 million spermatozoa = (proAKAP4 in ng/mL/spermatozoa M/mL) × 10 × 16 × (2/3). The 10× factor accounts for the final concentration in 10 million spermatozoa, while the 10× dilution factor corresponds to the post-thawed sample dilution during sample preparation. The 2/3 factor corresponds to the dilution volume loaded in each well of the 96-well plate in the Horse 4MID^®^ assay.

ProAKAP4 dosage was obtained from a replicate of loaded samples using Horse 4MID^®^ assays from the same batch. SPQI company provides the intraplate coefficient of variation below 10% on average and below 15% on average for an interplate of the same batch of between batches. ISO 13485 [30] accepts a variation coefficient below 15% as an quality control indicator.

### 2.5. Mitochondrial Membrane Potential

Mitochondrial membrane potential measurement was conducted utilizing the Guava^®^ Easycyte II™ flow cytometer (Guava Technologies Inc., Hayward, CA, USA, distributed by IMV Technologies, L’Aigle, France) and EasyKit™2 kit (IMV Technologies, L’Aigle, France) 37 of ready-to-use reagents (IMV Technologies, L’Aigle, France). The EasyKit™2 comprises a 96-well plate containing the JC-1 fluorochrome, which is specific for mitochondrial membrane potential.

At each time point (0, 1, and 3 h), 190 μL of EasyBuffer^®^ (IMV Technologies, L’Aigle, France), 10 μL of DMSO, and 1.75 μL of diluted semen were deposited in each well. Subsequently, the plate was incubated in darkness for 30 min at 37 °C. Analysis of the wells was performed using the Guava^®^ Easycyte II™ flow cytometer, calibrated beforehand with the Easy Check^®^ kit (Guava Technologies Inc., Massachussets, USA), and the Express Pro^®^ software (IMV, France).

The EasyKit™2 fluorochrome emits maximum fluorescence at 525 nm (green) and 596 nm (red). For each well, 5000 spermatozoa were analyzed. The JC-1 fluorescent stain serves as a marker of the intermediate piece of spermatozoa with high membrane potential and low membrane potential mitochondria. Polarized, active mitochondria are colored orange, while depolarized, inactive mitochondria are colored green, as per manufacturer-defined parameters.

Results are expressed as a percentage of “polarized” mitochondria (strongly or partially) and as a percentage of “depolarized” mitochondria. In our investigation, mitochondrial membrane potential is quantified by the percentage of “polarized” mitochondria, referred to as “active” mitochondria.

### 2.6. Statistical Analysis

The statistical analysis commenced with the assessment of data normality and homoscedasticity using the Shapiro–Wilk and Levene tests, respectively. In instances where data deviated from normal distribution, the arcsin √x transformation was employed to fulfill parametric assumptions. Subsequently, a mixed generalized linear model of repeated measures was applied to compare proAKAP4 concentration, total and progressive motility, and mitochondrial membrane potential of equine sperm across different incubation times (0 h, 1 h, and 3 h). Treatment (with a cryoprotectant and without a cryoprotectant) was considered as the fixed-effects factor, while individual horses were treated as random-effects factors. Furthermore, Pearson’s correlation analysis was conducted to assess the correlation coefficients (r) between ProAKAP4 concentration and sperm motility parameters, as well as mitochondrial membrane potential, within each treatment and at each time point.

Statistical analyses were conducted using the R statistical package (V 4.0.3, R Core Team; Vienna, Austria), while graphical representations were generated using GraphPad Prism software (V 8.4.0, GraphPad Software LLC; San Diego, CA, USA). The significance level was set at *p* < 0.05 for all analyses, and results are presented as the mean ± standard error of the mean (SEM). Analyses of the experimental data were performed with the statistical package R (V 4.0.3, R Core Team; Vienna, Austria), while the graphs were elaborated with the GraphPad Prism software (V 8.4.0, GraphPad Software LLC; San Diego, CA, USA). The minimum level of statistical significance was set at *p* < 0.05 for all analyses. Results are expressed as the mean ± standard error of the mean (SEM).

## 3. Results

### 3.1. Correlation between proAKAP4, Motility Descriptors and Mitochondrial Membrane Potential

Total and progressive motility showed a positive correlation with proAKAP4 concentration at 0, 1 and 3 h. However, considering the different descriptors of sperm motility, only those that refer to speed (VCL, VSL, VAP) and the sperm lateral head displacement (ALH) evidenced a positive correlation with proAKAP4 levels at 0 h (Table 1, Figure 1).

### 3.2. Evolution of proAKAP4, Motility Descriptors and Mitochondrial Membrane Potential over 3 h

No significant differences in proAKAP4 concentration were observed between the batch of samples with and without a cryoprotectant at 0 and 1 h of incubation. However, regardless of cryoprotectant removal, a significant decrease (*p* < 0.05) of proAKAP4 levels was observed at 3 h after thawing. The removal of the cryoprotectant had no effect on the concentration of proAKAP4 at 0 and 1 h, but showed a significant decrease (*p* < 0.05) at 3 h when the cryoprotectant was eliminated (Figure 2).

The evolution of total motility (MT) and progressive motility (MP) in percentages is shown in Figure 3A,B. In terms of total motility (MT), there was no difference between samples with and without a cryoprotectant and between incubation time. Regarding progressive motility (MP), no significant differences were observed between incubation time for booth type of samples, with or without a cryoprotectant. Nevertheless, samples with a cryoprotectant showed significantly lower progressive motility at all incubation times analyzed.

Analyzing the mitochondrial membrane potential, no significant difference was observed between samples (with or without a cryoprotectant) at different incubation times. However, in both cases, whether removing the cryoprotectant or not, a significant (*p* < 0.05) decrease between 0 and 1 h of incubation was observed but not between 1 and 3 h (Figure 4).

## 4. Discussion

The results of this experiment confirm a positive correlation between proAKAP4 concentration and total and progressive motility in stallion frozen/thawed semen at any time of our study. This positive correlation has been found previously in fresh semen in stallions [13,14,16] and in other mammals like bulls [23,31,32], jackass [24], dromedaries [28] rams [27], bucks [33], pigs [34], dogs [25], mice [35,36], and even in humans [37,38]. In the present study, this correlation between proAKAP4 and motility parameters is sustained during the 3 h post-thaw in contrast with the same experiment in donkeys [24] where this correlation between motility descriptors and proAKAP4 concentration was only described at time 0 and 1 h after thawing but not at 3 h. Horse and donkey, although phylogenetically close, are different species with their own reproductive strategies. All kinematic parameters for donkeys are significantly higher than for horse. Donkey sperm is more rapid and linear than horse one and, possibly, the effect of proAKAP4 on the motility patterns is also different between booth species [39]. Nevertheless, the obtained results confirm proAKAP4 as a good marker of sperm quality and longevity; kinematic parameters and longevity are both important for the fertilizing ability of frozen thawed spermatozoa [40,41]. Going deeper into the analytics of motility descriptors, ejaculates with a higher proportion of slow-moving spermatozoa demonstrate diminished levels of proAKAP4 [16]. Nonetheless, these findings underscore the relationship between proAKAP4 and sperm motility. A reduced proAKAP4 concentration may serve as an early indicator of semen deterioration over the time once the semen is thawed due to an elevated prevalence of subpopulation containing slowest spermatozoa. Within an ejaculate, this manifests as a decline in quality parameters such as viability or total motility. These findings on spermatic subpopulations complement prior research by Griffin et al. [14] who similarly reported a positive correlation between proAKAP4 concentration and the percentage of rapidly moving spermatozoa in ejaculates.

However, analyzing the descriptors of the sperm trajectory only the velocity parameters (VCL, VCL and VAP) and the lateral head displacement (ALH), and only at 1 h of post-thaw incubation but not later, correlate positively with proAKAP4 concentration. This correlation has been previously found in stallions [13] and in bulls [20,26,27,35]. In other mammals, like bulls, progressive motility [39] and velocity descriptors [41,42] have been described as sperm quality and even fertility markers. It seems evident that proAKAP4 has a relationship with the sperm motile ability but not an effect on the kind of movement, or trajectory description, in stallions [16]. Further studies are needed in order to evaluate the effect on frozen/thawed semen fertility.

ProAKAP4, as a reservoir of motility and energy, is described in several articles and drawn as a “backpack” with the energy of spermatozoa [12,18,34]. ProAKAP4 does not significantly decrease in the first hour but diminishes between one and three hours; in this case, there is no significant difference between the groups with or without cryoprotectants at 0 and 1 h. However, at 3 h, the samples without a cryoprotectant showed a significant decrease in the proAKAP4 concentration. On the other hand, on removing the cryoprotectant, higher levels of progressive motility were observed, indicating higher consumption of this energy “backpack”.

Total motility showed no significant difference between the centrifuged group (removing the cryoprotectant) and the non-centrifuged group (maintaining the cryoprotectant) and thus there is no non-significant difference between the 3 times studied. Centrifugation at 600 G during 10 min has not affect significantly total motility and viability parameters [43]. However, on removing the cryoprotectant, the progressive motility evidenced higher values at all analyzed times. This may indicate a deleterious effect of the cryoprotectant on sperm motility and, as explained before, higher proAKAP4 consumption when the progressive motility is higher. However, we cannot also rule out a certain effect of the viscosity of the medium when eliminating the cryoprotectant.

Comparing proAKAP4 with total and progressive motility, proAKAP4 holds well the first hour because it maintains motility and then suddenly decreases, but the motility has decreased slowly throughout the period. Comparing with data in donkeys [21], where our team carried out the same experiment, the results are very similar and comparable. It can be concluded that proAKAP4 is a structural protein that functions as the potential motility reserve of the equines spermatozoon [9] so by measuring the concentration it is probably possible to predict whether post-thaw motility will be constant over some time.

Longevity of post-thawed spermatozoa can be affected depending on the extender in different species. In deer, sperm longevity at four hours post-thawed is different depending on the cryoprotectant used [44]. In carp, supplement the extender with seminal plasma transferrin improves cryopreservation survival and longevity of the frozen-thawed semen [45]. In the present study, the used extender is an egg yolk-based extender (INRA Freeze ^®^). Egg yolk-based extenders are beneficial for semen preservation and post-thaw survival in tom cats [46] and significantly improve sperm thaw quality and longevity in African wild dog [47]. In stallions, cryoprotectant agents like phosphodiesterase inhibitors, caffeine, or taurine increase motility and longevity in post-thaw conditions [8].

No significant differences in pro AKAP4 concentration was observed in stallions between batches with or without a cryoprotectant; however, in donkey [24], a significant difference in proAKAP4 concentration has been found between the samples with or without a cryoprotectant. This difference can be explained because a different cryoprotectant was used in both experiments. Blommaert et al. [15] showed that proAKAP4 concentration differs significantly depending on the extender used in stallion thawed semen. Kowalczyk et al. [42] showed how the addition of Holothuroidea extract in the semen extender modulates positively the proAKAP4 concentration in the frozen/thawed bovine semen. In the present study, INRAFreeze^®^ (IMV Technologies, L’Aigle, France)was used to freeze all the ejaculates and in the case of donkey [24] BotuCrio^®^ (Nidacon, International AB, Mölndal, Sweden) was the election. Observed differences between donkey and horse can be explained because a different cryoprotectant has been used in both experiments; furthermore, other components could have an impact and there could be species-specific sensitivities, as explained before. Further analysis comparing these 2 extenders (BotuCrio^®^ and INRAFreeze^®^) is needed to elucidate if they exert the same effect on proAKAP4 or not. There are two hypotheses about these differences between extenders one is that a toxic effect might degrade proAKAP4 directly, and second, one extender might converse proAKAP4 more rapidly to AKAP4 than the other, dependent on the extender composition, depleting proAKAP4 supply which is no restored. Another way to address the lack of difference in batches with or without a cryoprotectant in our study could be the fact that centrifugation itself improves kinematic parameters and has a positive effect on proAKAP4 concentration [17], thus compensating for the potential toxic effect of cryoprotectant presence in the non-centrifuged batch.

For mitochondrial membrane potential, using the JC-1 stain, there is no significant difference between groups containing cryoprotectants or not, but the change over time is contrary to proAKAP4, decreasing rapidly between 0 and 1 h only. The absence of a relationship between the mitochondrial membrane potential and proAKAP4 levels suggests independence between spermatozoa energy production and the transmission of motion, On the other hand, mitochondria potential can be maintained in living/motile spermatozoa until their energy reserve is exhausted. This can explain why no direct relationship was observed in previous reports between mitochondrial activity stimulation and spermatozoa motility increase [48,49].

Further investigations are needed to compare the effect of the freeze–thaw cycle on proAKAP4 concentration and if this effect is the same for motility parameters and live–dead percentage of spermatozoa.

## 5. Conclusions

In conclusion, proAKAP4 is a good marker of sperm quality, correlating positively with total and progressive motility at any time in our study. It also correlates with velocity kinematic parameters at the moment of thawing. ProAKAP4 can be considered a long-lasting motility marker because it reduces at the time when motility parameters have decreased. Removing the cryoprotectant from frozen/thawed horse semen shows better maintenance of progressive sperm motility at 3 h and lower proAKAP4 concentration. Greater progressive motility means more proAKAP4 consumption.

There is no single analysis to evaluate semen quality. ProAKAP4 evaluation can be a complementary and valuable analysis. Combining these results with that of other semen quality markers in horses can help us to better predict stallion semen fertilizing ability depending on analysis practicality and economy. Nowadays, the use of CASA systems to evaluate sperm motility and flow cytometry is of great help. However, we cannot forget the analysis of seminal plasma in the quality control of semen.

## Figures and Tables

**Figure 1 animals-14-01264-f001:**
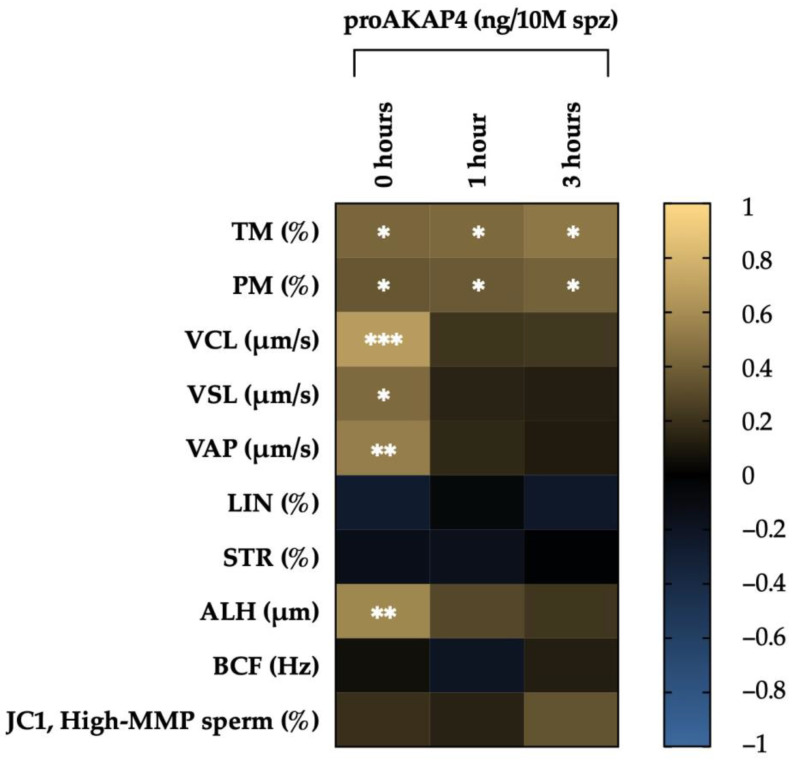
Graphical representation of correlations between sperm motility descriptors and proAKAP4 concentration. The colors of the scale (1 to −1) indicate whether the correlation is positive (yellow) or nehative (blue). * *p* ≤ 0.05, ** *p* ≤ 0.01, *** *p* ≤ 0.001.

**Figure 2 animals-14-01264-f002:**
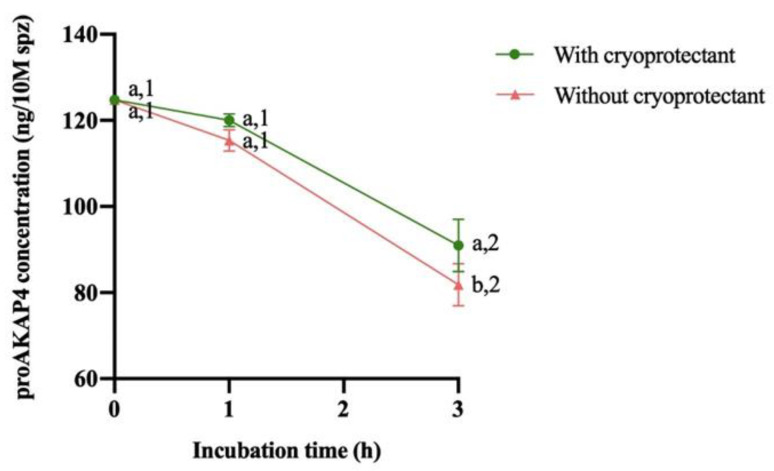
Change in proAKAP4 concentration after thawing and during the following 3 h of storage at 37 °C (mean ± SEM). Different numbers (1,2) mean significant proAKAP4 concentration differences (*p* < 0.05) between incubation times with or without a cryoprotectant. Different letters (a,b) mean significant proAKAP4 concentration differences (*p* < 0.05) at the same incubation time between treatment (with or without a cryoprotectant).

**Figure 3 animals-14-01264-f003:**
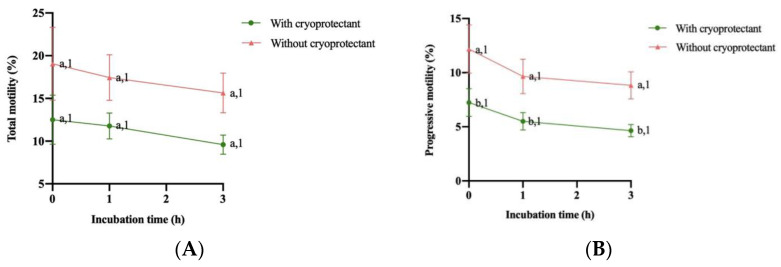
Change in (**A**) total motility (%) and (**B**) progressive motility (%) after thawing and during the following 3 h of storage at 37 °C (mean ± SEM). Different numbers (1,2) between mean significant differences (*p* < 0.05) between incubation time with or without a cryoprotectant. Different letters (a,b) show significant differences (*p* < 0.05) between treatment (with or without a cryoprotectant) at each incubation time.

**Figure 4 animals-14-01264-f004:**
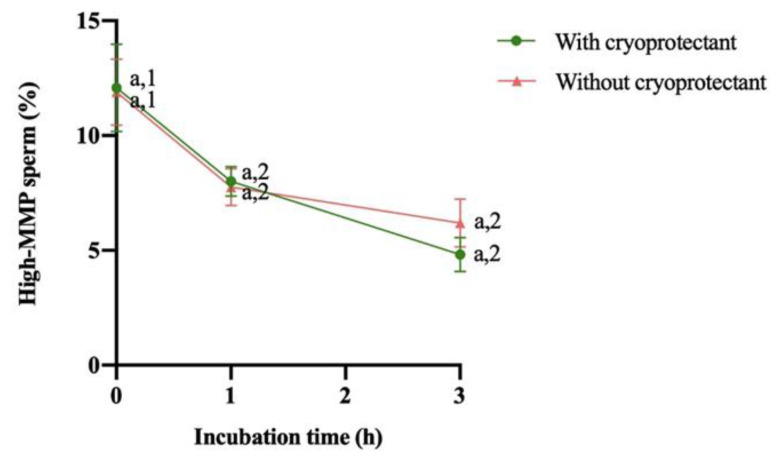
Change in mitochondrial membrane potential (MMP, %) after thawing and during the following 3 h of storage at 37 °C (mean ± SEM). Different numbers (1,2) mean significant differences (*p* < 0,05) between incubation time for the same treatment (with or without a cryoprotectant). No differences between treatment (with or without a cryoprotectant) were observed at the same incubation time (a).

**Table 1 animals-14-01264-t001:** Pearson’s correlation between proAKAP4 concentration in ng/10 M (millions) spermatozoa and motility descriptors and mitochondrial membrane potential at 0, 1 and 3 h after thawing. Total motility (TM, %), progressive motility (PM, %), curvilinear velocity (VCL, μm/s), straight-line velocity (VSL, μm/s), average path velocity (VAP, μm/s), linearity (LIN, %), straightness (STR, %), amplitude of lateral head displacement (ALH, μm), frequency of head displacement (BCF, Hz), and mitochondrial membrane potential (JC-1). * Mean significant correlation (*p* < 0.05). n = 64 ejaculates.

	0 h		1 h		3 h	
	r	*p*	r	*p*	r	*p*
TM	0.421	0.018 *	0.435	0.027 *	0.496	0.024 *
PM	0.352	0.041 *	0.367	0.039 *	0.402	0.027 *
VCL	0.666	0.000 *	0.211	0.247	0.222	0.223
VSL	0.440	0.012 *	0.139	0.449	0.120	0.511
VAP	0.527	0.002 *	0.160	0.382	0.105	0.567
LIN	−0.273	0.131	−0.069	0.707	−0.243	0.181
STR	−0.151	0.408	−0.168	0.359	−0.020	0.914
ALH	0.570	0.001 *	0.288	0.110	0.213	0.241
BCF	0.060	0.744	−0.205	0.260	0.122	0.505
JC-1	0.190	0.298	0.135	0.461	0.338	0.058

## Data Availability

Obtained data are included in this article.

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
