# Peer review of "ProAKAP4 as Indicator of Long-Lasting Motility Marker in Post-Thaw Conditions in Stallions"

_animals, 2024, doi:10.3390/ani14091264_

Round 1

Reviewer 1 Report

Comments and Suggestions for Authors

The authors have already studied ProAKAP4 as indicator of post-thaw motility in stallions, both in stallions and donkeys, this paper adds additional information regarding frozen semen from stallions by assessing the longevity of thawed semen. However, the title of the article does not reflect this additional information.

Therefore the title is not the most appropriate for this article "ProAKAP4 as indicator of post-thaw motility in stallion" when the conclusion of the authors in their 2022 article concluded with the following sentence: “Concentrations of proAKAP4 higher than 37.77 ng/10M spz were correlated with a very good quality frozen/thawed stallion semen”.

In the discussion:

There is no reference or discussion of the values obtained in the parameters evaluated. There are reference values for both motility (line 55) and its characteristics such as ProAKAP4 (lines 413-415), it would be interesting to discuss the results obtained in this study.

In lines 278-286, they refer to articles that evaluate the relationship of ProAKAP4 with motility characteristics, but do not make any reference to the results of 2022 where in addition to motility characteristics they evaluated sperm subpopulations. It would be very interesting to go deeper into these aspects.

Likewise, the discussion should go deeper into the novelties that this article reflects, such as the study of longevity, and the presence or absence of cryoprotectants in the parameters evaluated.

Author Response

Dear reviewer, thank you for your kind review that help us to improve the article.

General Comments

The authors have already studied ProAKAP4 as indicator of post-thaw motility in stallions, both in stallions and donkeys, this paper adds additional information regarding frozen semen from stallions by assessing the longevity of thawed semen. However, the title of the article does not reflect this additional information.

Therefore the title is not the most appropriate for this article "ProAKAP4 as indicator of post-thaw motility in stallion" when the conclusion of the authors in their 2022 article concluded with the following sentence: “Concentrations of proAKAP4 higher than 37.77 ng/10M spz were correlated with a very good quality frozen/thawed stallion semen”.

ANSWER

The title has been changed as required 

Discussion

There is no reference or discussion of the values obtained in the parameters evaluated. There are reference values for both motility (line 55) and its characteristics such as ProAKAP4 (lines 413-415), it would be interesting to discuss the results obtained in this study.

ANSWER

Obtained results are deeper discussed in the new version.

In lines 278-286, they refer to articles that evaluate the relationship of ProAKAP4 with motility characteristics, but do not make any reference to the results of 2022 where in addition to motility characteristics they evaluated sperm subpopulations. It would be very interesting to go deeper into these aspects.

ANSWER

A reference has been included

Likewise, the discussion should go deeper into the novelties that this article reflects, such as the study of longevity, and the presence or absence of cryoprotectants in the parameters evaluated.

ANSWER

The discussion was modified according to your suggestion

Reviewer 2 Report

Comments and Suggestions for Authors

Introduction

A large part of the introduction deals with generalities about insemination and sperm quality parameters in stallions. But this is not the focus of the study. More details on the interest of using ProAKP4 as an indicator of sperm quality could be added. Including human and horse data. How is this indicator complementary to the other tests already available? And how it could help to solve the problem of the lack of correlation between most tests and the fertilizing ability of frozen/thawed semen? The authors do not mention either how this study is complementary to other studies testing ProAKP4 as a “new” marker of semen quality in the horse?

Material and methods

L94: Does it mean 4 different ejaculates by stallion? Please be more precise.

It seems that the proportion of dead sperm cells was not measured? ProAKP4 concentrations are low in dead semen and so the proportion of dead sperm cells affect the mean ProAKP4 concentration of the sample. It is well known that the proportion of dead sperm cells is much higher after cryopreservation than in fresh semen, and increases steadily after thawing. It is highly variable between samples, stallions, extenders, temperature post-thawing... Thus if there are variations in the proportion of dead sperm cells between samples and between time points (as expected), it will also affect the interpretation of the ProAKP4 data. Dead sperm cells are not taken into account for evaluating the motility parameters like VCL, ALH etc… as they are immotile. On the contrary, dead/immotile sperm cells mitochondria will show green fluorescence with JC1 staining so this staining takes into account the proportion of dead/immotile spz.

L 117 The way semen concentration was evaluated by CASA is not specified. It is an important parameter to accurately evaluate ProAKP4 concentrations. Were the results reported to all sperm cells or only the living/motile ones?

L. 130 How many sperm cells were evaluated per sample?

L153 Please add information about the spectrophotometer used.

L157 Do you have indications about the reproducibility of the test (intra- and inter-assay variability)?

Statistical analysis:

Was each ejaculate analyzed only once?

Results

For each result, please mention the number of data and how was taken into account the fact that the 16 samples were coming from 4 stallions.

Correlations and table 1: please mention how many data were used for the correlation tests. How was the variability within and between stallions? A graphical presentation of some of the significant correlations will be useful. It is well known that a p value for a correlation is not informative enough. Moreover, the highest square r coefficient of determination is not very high (0.44 for VCL at 0h).

Fig 2 and discussion L292, L299-302 The difference for the motility parameters with and without cryoprotectants is likely to be due to the difference in viscosity of the medium. Indeed, a lower viscosity after removing the cryoprotectants will increase the proportion of motile and progressive spz if the thresholds are not changed. This is also observed between different cryoprotectants if the viscosity is not the same.

Discussion

As already mentioned, the proportion of dead sperm cells is high after freezing-thawing, varies between ejaculates and stallions. It is important to discuss the point.

The differences with and without cryoprotectants on the motility parameters need to be related to the differences in viscosity (see supra). This is supported by the fact that no difference is observed for the JC1 staining.

L323 and L331 : same sentence repeated twice

L334 The difference in ProAKP4 concentrations between extenders can also be correlated with the sperm cells survival…

L340-342 : this hypothesis should be nuanced. Indeed, once the sperm cells recover their motility after thawing, ProAKP4 will rapidly be activated into AKP4, so depleting ProAKP4 supply which is not restored. On the other hand, mitochondria potential will be maintained in living/motile sperm cells until their energy reserve is exhausted.

Conclusion

As already mentioned, it is important to take into account the dead sperm cells when interpreting the results. The conclusion should be modified based on the remarks made above. Based on the study and the literature it seems that ProAKP4 evaluation will not help to better predict stallion semen fertilizing ability but could maybe replace some other analysis. Could the authors comment about that? What would be the best (hypothetical) combination of markers of semen quality in the horse?

Author Response

Dear reviewer,

Thank you for your accurate review that help us to improve the manuscript.

Introduction

A large part of the introduction deals with generalities about insemination and sperm quality parameters in stallions. But this is not the focus of the study. More details on the interest of using ProAKP4 as an indicator of sperm quality could be added. Including human and horse data. How is this indicator complementary to the other tests already available? And how it could help to solve the problem of the lack of correlation between most tests and the fertilizing ability of frozen/thawed semen? The authors do not mention either how this study is complementary to other studies testing ProAKP4 as a “new” marker of semen quality in the horse?

ANSWER

As you suggest we introduced how our study is complementary to other studies testing PrtoAKAP4.

Material and methods

L94: Does it mean 4 different ejaculates by stallion? Please be more precise.

ANSWER

It has been clarified

It seems that the proportion of dead sperm cells was not measured? ProAKP4 concentrations are low in dead semen and so the proportion of dead sperm cells affect the mean ProAKP4 concentration of the sample. It is well known that the proportion of dead sperm cells is much higher after cryopreservation than in fresh semen, and increases steadily after thawing. It is highly variable between samples, stallions, extenders, temperature post-thawing... Thus if there are variations in the proportion of dead sperm cells between samples and between time points (as expected), it will also affect the interpretation of the ProAKP4 data. Dead sperm cells are not taken into account for evaluating the motility parameters like VCL, ALH etc… as they are immotile. On the contrary, dead/immotile sperm cells mitochondria will show green fluorescence with JC1 staining so this staining takes into account the proportion of dead/immotile spz.

ANSWER

proAKAP4 has been demonstrated that is a motility marker then we put the focus to the motile or not motile cells, and dead cells are considered in the not motile group.

 L 117 The way semen concentration was evaluated by CASA is not specified. It is an important parameter to accurately evaluate ProAKP4 concentrations. Were the results reported to all sperm cells or only the living/motile ones?

ANSWER

As explained before the results take into account the living/motile spermatozoa.

L. 130 How many sperm cells were evaluated per sample?

ANSWER

At least 1000 spermatozoa. In cluded in the text.

L153 Please add information about the spectrophotometer used.

ANSWER

It has been included.

L157 Do you have indications about the reproducibility of the test (intra- and inter-assay variability)?

ANSWER

The test used was a kit from spqi-4biodx, wich is a CE registered company and its kid was tested before comercialization.

Statistical analysis:

Was each ejaculate analyzed only once?

3 times for the CASA and once for proAKAP4

Results

For each result, please mention the number of data and how was taken into account the fact that the 16 samples were coming from 4 stallions.

ANSWER

N has been indicated. There are a variation between ejaculates and each ejaculate was taking as an individual sample. 

Correlations and table 1: please mention how many data were used for the correlation tests. How was the variability within and between stallions? A graphical presentation of some of the significant correlations will be useful. It is well known that a p value for a correlation is not informative enough. Moreover, the highest square r coefficient of determination is not very high (0.44 for VCL at 0h).

ANSWER

Data number was included in Table 1 and a graphical representation was added (Figure 1).

 Fig 2 and discussion L292, L299-302 The difference for the motility parameters with and without cryoprotectants is likely to be due to the difference in viscosity of the medium. Indeed, a lower viscosity after removing the cryoprotectants will increase the proportion of motile and progressive spz if the thresholds are not changed. This is also observed between different cryoprotectants if the viscosity is not the same.

ANSWER

Changed at the discussion

Discussion

As already mentioned, the proportion of dead sperm cells is high after freezing-thawing, varies between ejaculates and stallions. It is important to discuss the point.

ANSWER

Included.

The differences with and without cryoprotectants on the motility parameters need to be related to the differences in viscosity (see supra). This is supported by the fact that no difference is observed for the JC1 staining.

ANSWER

Thanks for your interesting hipothesis. A possible effect of the viscosity was included in the text of the reviewed mnanuscript.

L323 and L331 : same sentence repeated twice

ANSWER

It has been corrected.

L334 The difference in ProAKP4 concentrations between extenders can also be correlated with the sperm cells survival…

ANSWER

As explained before, proAKAP4 is a motility marker then we put the focus to the motile or not motile cells. Dead spermatozoa ae included in the non motile group.

L340-342 : this hypothesis should be nuanced. Indeed, once the sperm cells recover their motility after thawing, ProAKP4 will rapidly be activated into AKP4, so depleting ProAKP4 supply which is not restored. On the other hand, mitochondria potential will be maintained in living/motile sperm cells until their energy reserve is exhausted.

ANSWER

Thanks for your comment that was considered in the new version.

Conclusion

As already mentioned, it is important to take into account the dead sperm cells when interpreting the results. The conclusion should be modified based on the remarks made above. Based on the study and the literature it seems that ProAKP4 evaluation will not help to better predict stallion semen fertilizing ability but could maybe replace some other analysis. Could the authors comment about that? What would be the best (hypothetical) combination of markers of semen quality in the horse?

ANSWER

A new paragraph was included in our discussion taking into account your comments.

Round 2

Reviewer 1 Report

Comments and Suggestions for Authors

The authors have successfully addressed all the comments, they have modified the title to better match the content of the article, expanded the discussion and improved the manuscript in the new version.

I recommend that this revised version be accepted for publication.

Author Response

Dera Editor,

Thank you very much for your kind review.

Reviewer 2 Report

Comments and Suggestions for Authors

see attached file

Author Response

Dear reviewer thanks for you kind comments.

Comments on the revised version
Introduction: some information has been added but the authors did not mention why ProAKP4 could be complementary to other motility markers and the reason why it could be interesting to measure this parameter post-thawing (see also discussion).

ANSWER

Some changes are introduced in the introduction  and discussion in the new version of the article taking into acount your comments.

Comments and cites on proAKAP4 relationship with sperm viability and its usefulness as a marker of sperm motility are included.

Results and statistical analysis:
the authors now mentioned that 4 ejaculates of each stallion were analyzed… but for the correlations, it seems that they consider all samples as independent from each other, while for the comparisons between means, they used the correct analysis (mixed analysis) taking into account the stallion effect.
The added fig 1 does not represent what was asked: a correlation graph showing for at least one parameter the correlation between pro-AKP4 measured on each sample and a measured quality parameter (like ALH): 16 data points showing also which points correspond to which stallion).
For those reasons I am not yet convinced by the credibility of the correlation data

ANSWER

Initially, the sole purpose of the correlations was to corroborate the consumption of proAKAP reflected in motility over time, as has already been demonstrated in previous studies (represented in Fig1)

On the other hand, the goal was to look at what happens to the concentration of proAKAP4 in each treatment (with and without cryoprotectant) as a function of time. The goal is not correlations.

We add as supplementary files the correlation between proAKAP4 at diferrent times (0,1,3h).

The authors were asked to take into account the dead sperm cells in the interpretation of the data as for some parameters the dead sperm cells are included and for others not (which depends on the parameter: no motility data for dead sperm, but ProAKP4 or JC1 data include dead spermatozoa).
This is not yet clear.

ANSWER

proAKAP4 is not correlated with viability as not all viable spermatozoa are stained by proAKAP4 in stallion sperm after thawing (Blommaert et al., 2018). ProAKAP4 is detected in living spermatozoa but with cell death proAKAP4 is degraded (Nixon et al., 2019) (included in the introduction).

The authors mention that no data on intra- and inter-assay seem available for the ProAKP4 kit. Was therefore an internal control included in each replicate?

ANSWER

ProAKAP4 dosage in the present article was obtained from a replicate of loaded samples using Horse 4MID® assays from the same batch. SPQI company provides the intraplate coefficient of variation below 10% on average and below 15% on average for interplate of the same batch of between batches. ISO 13485 accepts a variation coefficient below 15% as an quality control indicator (included in M&M).

Discussion
Some comments/suggestions were taken into account but the general conclusion about the interest to add ProAKP4 as a marker complementary to other motility parameters is not developed. ProAKP4 concentration measure the reserve in AKP4, a kinase that is essential for sperm motility. This protein
is expressed and accumulated during spermatogenesis before sperm nucleus condensation. It gives thus an information independent of energy stores. The authors should go more in depth in the discussion on this point.

ANSWER

As commented before, some changes are introduced in the introduction and discussion taking into account your suggestions.